# Efficient Methodology for Detection and Classification of Short-Circuit Faults in Distribution Systems with Distributed Generation

**DOI:** 10.3390/s22239418

**Published:** 2022-12-02

**Authors:** Andréia da Silva Santos, Lucas Teles Faria, Mara Lúcia M. Lopes, Anna Diva P. Lotufo, Carlos R. Minussi

**Affiliations:** 1Department of Electrical Engineering, São Paulo State University (UNESP), Ilha Solteira 15385-000, SP, Brazil; 2Department of Energy Engineering, São Paulo State University (UNESP), Rosana 19273-000, SP, Brazil

**Keywords:** distribution systems, distributed generation, fuzzy logic inference, wavelet transform, multi-resolution analysis, short-circuit fault detection, short-circuit fault classification

## Abstract

Fault detection and classification are crucial procedures for electric power distribution systems because they can minimize the occurrence of faults. The methods for fault detection and classification have become more problematic because of the significant expansion of distributed energy resources in distribution systems and the change in their currents due to the action of short-circuiting. In this context, to fill this gap, this study presents a robust methodology for short-circuit fault detection and classification with the insertion of distributed generation units. The proposal methodology progresses in two stages: in the former stage, the detection is based on the continuous analysis of three-phase currents, whose characteristics are extracted through maximal overlap discrete wavelet transform. In the latter stage, the classification is based on three fuzzy inference systems to identify the phases with disturbance. The short-circuit type is identified by counting the shorted phases. The algorithm for short-circuit fault detection and classification is developed in MATLAB programming environment. The methodology is implemented in a modified IEEE 34-bus test system and modeled in ATPDraw with three scenarios with and without distributed generation units and considering the following parameters: fault type (single-phase, two-phase, and three-phase), angle of incidence, fault resistance (high impedance fault and low impedance fault), fault location bus, and distributed generation units (synchronous generators and photovoltaic panels). The accuracy is greater than 94.9% for the detection and classification of short-circuit faults for more than 20,000 simulated cases.

## 1. Introduction

Electric power distribution systems (EPDS) have the continuous supply of electric energy to consumer units as their main objective. However, these systems are exposed to several factors that can cause disturbances, steady-state faults, or blackouts. The faults are classified as temporary or permanent. Temporary faults cause a transient disturbance in EPDS, which automatically returns to its normal operating state (NOS). On the other hand, permanent faults cause an interruption in the energy supply; thus, EPDS will return to its NOS after the short-circuit repair. The faults are characterized by impermissible deviation from the standard operating conditions of EPDS [1].

Distribution systems have changed with the massive insertion of distributed energy resources (DERs) based on renewable energy [2]. DERs expansion is essential to meet the growing energy demand. In addition, environmental concerns have been necessary to encourage renewable energy expansion. The increase in renewable distributed generation (DG) units reduce greenhouse gases [3].

On the other hand, the arbitrary and massive insertion of DG units into EPDS causes significant changes in its configurations. The electrical system loses its radial characteristic as multiple power sources power it. The main impact of these changes is related to the conventional protection system, since features such as direction, amplitude, and short-circuit currents are changed [3,4].

### 1.1. Literature Review

Several studies in the specialized literature developed tools for fault analysis in EPDS by inserting DG units. They addressed the impact on the conventional protection system due to the significant growth of DG units into EPDS. The following studies discussed different methodologies for fault detection and classification in EPDS.

Rai et al. [5] proposed a convolutional neural network technique for fault classification in EPDS with DG units. The proposed method does not have the pre-processing phase; thus, three-phase currents and voltage signals were applied directly as inputs.

A new method for short-circuit faults detection and classification in balanced and unbalanced distribution systems was presented by Zhang et al. [6]. The current’s positive, negative, and zero sequence components characterized the faults. The operating modes of DG units were considered using the Fortescue approach. A softmax regression model was introduced to minimize the impact of transient signals on the fault classification module and applied only two resistors with maximum value during the simulations.

The discrete wavelet transform (DWT) was used in [7] on three-phase current signals, whose values were applied to formulate a decision tree to perform the fault classification. Decision tree input was formed by a comparison parameter obtained using the maximum value of three-phase current signals and zero-sequence. Those parameters were applied as a reference to detect the fault phases. The fault resistance variation was not considered.

In [8], a strategy based on fuzzy logic was used for fault detection and classification. Chaitanya et al. implemented two fuzzy inference systems (FIS). They were built to detect and classify low-impedance faults (LIF) and high-impedance faults (HIF). A Teager Energy Operator (TEO) was applied to extract three-phase current signal characteristics, which composed the FIS input set.

In [9], fault detection was performed by applying the power spectral density calculated from the wavelet covariance matrix. The signal information was extracted through a wavelet transform (WT), where the signals were decomposed into three levels using the *db4* mother wavelet.

Elnozahy et al. [10] proposed a new method for detecting and classifying single-phase faults using DWT and artificial neural networks (ANN). The transient signals were analyzed based on the *db4* mother wavelet to extract their characteristics.

A method based on DWT and ANN for HIF detection was presented by Silva et al. [11]. The authors focused on incremental learning procedures to find new fault patterns. DG units were not considered.

Decanini et al. [12] presented a method for automatic fault diagnosis. The detection process was based on statistical and direct analysis of three-phase currents and voltage signals in the wavelet domain. DWT and multi-resolution analysis (MRA) were introduced to extract the signal characteristics. The short-phase classification was performed by a set of Fuzzy-ARTMAP ANN. DG units were not considered in the system modeling.

A combination of maximum overlap discrete wavelet packet transform (MODWPT) and empirical mode decomposition (EMD) were applied for HIF detection in [13]. That methodology was based on the estimation of fault current signals inter-harmonic through MODWPT.

In [14], a method based on an adaptive neural fuzzy inference system was proposed for fault classification. Three-phase currents and voltage signals measured at the substation output were evaluated. The signal transient components were extracted via WT.

Different fault types were detected and classified in [15] using a combination of wavelet singular entropy theory (WSE) and fuzzy logic. The algorithm was based on the singular wavelet values of each phase, since a phase with an anomaly presents values outside of the allowed limit.

A proposal based on deep learning and deep belief networks (DBN) was implemented by Hong et al. [16] for fault classification in EPDS with and without DG units. After the DBN training with current and voltage signals, it was possible to obtain the signal characteristics and classify the fault types that occurred in EPDS.

Ola et al. [17] presented an algorithm for fault classification, based on the fault index. It was calculated from the three-phase current signals, in which the Wigner distribution function and the decomposition based on the coefficient of alienation were applied. The anomaly phases were identified from the fault index. The fault classification was carried out from the count of phases with anomaly.

A methodology for HIF detecting was presented in [18]. The technique was based on EMD with adaptive noise applied to decompose the zero-sequence current signals to obtain the intrinsic mode function (IMF). TEO was applied to identify changes in characteristics of IMF waveforms. HIF identification was performed by calculating the signal WSE. WSE for faulty signals are higher than that one for normal signals.

Sekar et al. [19] proposed an intelligent method based on WT and data mining for HIF detection. The main characteristics of three-phase current signals were extracted via WT.

In [20], the authors presented a technique for identifying HIF. A method based in Adaline ANN was applied to extract the third harmonic angle (THA) from current signals. Adversarial generative conditional was applied to produce input data from THA. Finally, a convolutional neural network was applied to separate HIF from others transient events.

Wontroba et al. [21] proposed a methodology based on the analysis of harmonic and symmetrical components of currents to detect HIF due to cable break. HIF detection process has two steps: (i) cable break detector and (ii) HIF detector. In the former step, the identification of a cable break was carried out by applying the phase current phasors. In the latter step, HIF detection was carried out via application of neutral current and analysis of its harmonic components.

Yuan et al. [22] presented a method for detecting faulty feeders after a single line-to-ground fault (SLGF). It was based on waveform recognition, where zero-sequence voltage (ZSV) on bus and local zero-sequence current (ZSC) were applied. An image for each feeder was constructed via ZSV and ZSC superposition. Finally, a convolutional neural network was applied to perform image recognition, where all fault features were extracted adaptively.

### 1.2. Contributions

This study presents a robust methodology for fault detection and classification in EPDS with DG units. The highlights of the proposed approach are described below.

Development of a simple and effective methodology for short-circuit faults detection and classification of distribution systems that included the DG units.Sensitivity analysis to assess the impacts on fault detection and classification in each proposed scenario. There are synchronous generators and photovoltaic (PV) panels with different levels of DG units’ insertions into EPDS.Maximum overlap discrete wavelet transforms (MODWT) and FIS for faults detection and classification, respectively.

### 1.3. Paper Structure

The rest of this paper is as follows: Section 2 explains the techniques used in short-circuit fault detection: problem description (Section 2.1), HIF (Section 2.2), MODWT (Section 2.3), the steps for fault detection (Section 2.4), and the confusion matrix metrics (Section 2.5). Section 3 addresses the methodology used for short-circuit fault classification: a FIS for short-circuit faults classification (Section 3.1), FIS rules set (Section 3.2), and the steps for short-circuit faults classification (Section 3.3). Section 4 presents the results and discussions. All simulated cases are applied in the modified IEEE-34 bus test system addressed in Section 4.1. The results for detection and classification of short-circuit faults and the validation for both are explained and discussed in Section 4.2, Section 4.3 and Section 4.4, respectively. Finally, Section 5 presents the conclusions.

## 2. Short-Circuit Fault Detection

### 2.1. Problem Description

Energy supply interruptions occur due to several factors such as: equipment failures, atmospheric discharges, human errors, contact of trees with the overhead distribution network, and fires in the vicinity of the electrical network.

EPDS are susceptible to four fault types: line-to-ground fault (SLGF), double line-to-ground fault (DLGF), line-to-line fault (LLF), and three-phase-ground fault (LLLGF). SLGF is the most frequent fault type, with 70% of classified cases [1]. Figure 1 shows an example to illustrate the fault types, where there are 12 cycles (200 ms) of three-phase currents with a frequency of 60 Hz. Figure 1a shows an SLGF fault, where a short-circuit is applied at t=66 ms (4th cycle) in phase C with ground: $C_{g}$. After the disturbance, the fault phase current has its value increased. Thus, it is easy to note the difference in current magnitude between the normal phases A and B with respect to phase C where there is an anomaly. Figure 1b shows a DLGF fault where the phases A and B are short-circuited: $AB_{g}$. We observe that the magnitude of phase C current in NOS is lower than that ones in phases A and B, because they have anomalies. Finally, Figure 1c shows an LLLGF fault. At t=66 ms (4th cycle) a three-phase short-circuit is applied, which causes an increase in the current magnitude after the disturbance.

### 2.2. High-Impedance Faults

HIFs are caused by direct contact of electrical wires from the distribution network with high-impedance objects or surfaces such as concrete, sand, asphalt roads, and tree branches. They are characterized by low magnitudes of fault currents, which may be insufficient to excite an overcurrent relay. Thus, fault detection becomes very difficult, since the conventional protection devices are based on overcurrent. During HIF, the short-circuit current presents values equivalent to the rated current of the electrical system at the fault location [11,23]. On the other hand, LIFs can be detected easily by established methodologies in specialized literature.

In this study, we apply a HIF model with two diodes in parallel [23]. As shown in Figure 2, the model is composed of two DC sources, $V_{p}$, and $V_{n}$, and two resistors, $R_{p}$ and $R_{n}$, that represent the fault resistances.

### 2.3. Maximum Overlap Discrete Wavelet Transform

A MODWT performs a linear filtering operation, where a series is transformed into coefficients associated with variations over a scale set. It was developed to act similarly to DWT [24]. Unlike DWT, MODWT can be applied to all sample sizes and it has flexibility in the initial point selection, where the coefficients do not change as a result of changes in the initial point [25].

The filtering process consists of applying a high-pass wavelet filter at a given decomposition level j to produce a set of wavelet coefficients and a low-pass scaling filter to produce a set of scaling coefficients [26]. After the filtering process, the data size becomes twice the size of original signal.

On the other hand, DWT is applied in a simplified filtering process. This process is performed through the down-sampling operator whose objective is to reduce the number of samples at the filter output. MODWT does not apply the down-sampling operator; thus, at each level j of decomposition, the wavelet coefficients will have the same length as the original signal as shown in Figure 3 [26,27].

The MODWT scale filter ${\tilde{h}}_{j,\ell }$ and the wavelet filters ${\tilde{g}}_{j,\ell }$ are defined in (1) and (2) based on the DWT filters $g_{j,\ell }$ (low-pass) and $h_{j,\ell }$ (high-pass) [24,25,26,27]:(1)$${\tilde{h}}_{j,\ell }=\;\frac{h_{j,\ell }}{\sqrt{2}}$$
(2)$${\tilde{g}}_{j,\ell }=\;\frac{g_{j,\ell }}{\sqrt{2}}$$
where ℓ=0,…,L−1 and L represents the filter length. The subscript j is the level to which the signal will be decomposed.

For a given level j, the MODWT coefficients can be defined as the convolution of the time series $X_{n}$ according to (3) and (4) [25]:(3)$${\tilde{W}}_{j,n}=\;\overset{L-1}{\underset{\ell =0}{\sum }}{\tilde{h}}_{j,\ell }X_{n-\ell }\;\;mod\;N$$
(4)$${\tilde{V}}_{j,n}=\;\overset{L-1}{\underset{\ell =0}{\sum }}{\tilde{g}}_{j,\ell }X_{n-\ell }\;\;mod\;N$$

By means of (5) and (6), we obtain for the nth signal sample at the jth decomposition level the approximation and detail components ${\tilde{A}}_{j}$ and ${\tilde{D}}_{j}$ respectively [25]. Where ${\tilde{g}}_{\ell }^{0}$ and ${\tilde{h}}_{\ell }^{0}$ are periodized of $\tilde{g}$ and $\tilde{h}$ to length N, respectively. N corresponds to the signal sample size:(5)$${\tilde{A}}_{j,n}=\;\overset{L-1}{\underset{\ell =0}{\sum }}{\tilde{g}}_{j,\ell }^{0}{\tilde{V}}_{j,\;\;n+\ell }\;\;mod\;N$$
(6)$${\tilde{D}}_{j,n}=\;\overset{L-1}{\underset{\ell =0}{\sum }}{\tilde{h}}_{j,\ell \;}^{0}{\tilde{W}}_{j,\;\;n+\ell }\;\;mod\;N$$

Figure 3 shows a filtering process via MODWT, in which the original signal is decomposed into three levels of resolution. At each level j, the approximation and detail coefficients $A_{j}$ and $D_{j}$ are the same size as the original signal, because there is a retention of reduced values at each level of decomposition which are normally discarded by the down-sampling operator applied in DWT. Therefore, these values are preserved in MODWT [22].

### 2.4. Fault Detection Methodology

The fault detection is based on the real-time monitoring of three-phase current signals measured at the substation output. The signal sample characteristics are extracted via MODWT, where the detail and approximation coefficients are obtained from signal decomposition. Signals are decomposed into three levels of resolution by applying a fourth-order filter from Daubechies family: db4. There are high and low-frequency signals in detail and approximation coefficients.

NOS of electrical system is defined via normalized energy of detail coefficients at the third level and by means of current signals. The difference between NOS and a fault condition is determined via a comparative analysis of energies and a normalized mean with a previously established threshold of variation for each phase. If the normalized energy or normalized mean of at least one phase exceeds the threshold, then an anomaly or fault was identified in the electrical system.

The magnitudes of faulted phase currents are significantly higher than the ones in NOS as shown in Figure 1. Thus, the current means that belong to defective phases will be greater than the ones that belong to normal phases.

The algorithm for faulted phase detection is divided into five steps: (i) signal decomposition; (ii) energy signal calculation; (iii) three-phase currents mean; (iv) normalization; and (v) faulted phase detection.

#### 2.4.1. Step 1: Signal Decomposition

The data window consists of one post-fault cycle, and it is decomposed as follows:
Data window is decomposed into three resolution levels j via MODWT using *db4* mother wavelet of Daubechies family.Applying the fourth order filter of Daubechies family, we obtain the detail coefficients with three resolution levels.

#### 2.4.2. Step 2: Signal Energy

The energy of current signals for each phase is obtained via (7) according to the Parseval’s theorem [29]:(7)$$E_{i,j}=\overset{N}{\underset{n=1}{\sum }}{\left|d_{i,j}^{n}\right|}^{2}\;with\;j=3$$
where $E_{i,j}$ is the energy for the detail coefficients to phase i current signal and $d_{i,j}^{n}$ represents the nth sample of detail coefficients for the phase i at the third level of signal decomposition—j=3.

Figure 4 shows an illustrative example of normalized energies obtained for all fault types considered in this study. The energies values of phases with a disturbance present are greater than phases without anomaly.

The energy vector is shown in (8). It is composed by energies of $I_{a}$, $I_{b}$, and $I_{c}$ three-phase currents:(8)$$E=\left[\begin{array}{ccc} \;E_{a,j} & \;E_{b,j} & \;E_{c,j} \end{array}\right]\;\mathrm{with}\;j=3$$

#### 2.4.3. Step 3: Three-Phase Currents Mean

The currents mean of each phase are obtained via (9) from a data window composed of three cycles with 180 samples (one cycle) extracted from post-fault current signals. The vector of three-phase currents mean is shown in (10). Figure 5 shows an illustrative example, where the defective phases present a higher mean of normalized current signal than that ones for normal phases.
(9)$$M_{i}=\;\frac{\sum M_{i}}{180},\;\mathrm{with}\;i\in \left\{a,b,c\right\}$$
(10)$$M=\left[\;\begin{array}{ccc} M_{a} & M_{b} & M_{c} \end{array}\;\right]$$

#### 2.4.4. Step 4: Normalization

The normalization ${\bar{M}}_{i}$ and the mean of energy signals ${\bar{E}}_{i}$ are performed via (11) and (12), respectively.
(11)$${\bar{M}}_{i\;}=\frac{M_{i}}{\max\left(M\right)},\;\mathrm{with}\;i\in \left\{a,b,c\right\}\;$$
(12)$${\bar{E}}_{i\;}=\frac{E_{i,j}}{\max\left(E\right)}\;\mathrm{with}\;j=3$$

#### 2.4.5. Step 5: Detection

The system operation state is defined via (13):(13)$$\mathrm{Electrical}\;\mathrm{System}=\{\begin{array}{ll} \mathrm{With}\;\mathrm{Anomaly}, & \mathrm{if}\;\;Max\left[{\bar{E}}_{\;}\right]>LD_{i}\;\mathrm{and}\;Max\left[\bar{M}\right]>LD_{i}\\ \mathrm{Without}\;\mathrm{Anomaly}, & \mathrm{if}\;\;\;Max\left[{\bar{E}}_{\;}\right]\leq LD_{i}\;\mathrm{or}\;Max\left[\bar{M}\right]\leq LD_{i} \end{array}$$
where the Max[·] function returns the maximum value in an array. The limit for detection ($LD_{i}$) is that one allowed for the phase i current.

### 2.5. Detection Validation

In this section, the well-known confusion matrix and the metrics obtained from its elements are presented to evaluate the efficiency of the short-circuit fault detection stage.

#### 2.5.1. Confusion Matrix

The confusion matrix metrics are applied to evaluate the detection stage performance. It is worth pointing out that the confusion matrix application is feasible because the output of the detection stage is binary (absence or presence of short-circuit faults).

Table 1 shows the structure of confusion matrix adapted for this study.

Follow below a description of confusion matrix elements:
$Q_{NN}$: number of normal cases (without fault) and whose detection was correct (true-positive cases);$Q_{NF}$: number of normal cases (without fault); however, they were incorrectly detected as fault (false-negative cases);$Q_{FN}$: number of fault cases; however, they were incorrectly detected as normal (without fault)—false-positive cases;$Q_{FF}$: number of fault cases and whose detection was correct (true-negative cases).

#### 2.5.2. Confusion Matrix Metrics

In this section, the accuracy, precision, recall, and f1-score metrics obtained from the confusion matrix are presented to evaluate the short-circuit detection stage.

Accuracy in (14) is the ratio between the number of hits (true-positive and true-negative cases) and the number of all simulated cases. This metric alone is insufficient for evaluating problems where there is an imbalance between the simulated cases (normal and fault cases). Therefore, other metrics such as precision, recall and f1-score are evaluated together with the accuracy to support the evaluation of the short circuit detection stage proposed in this study.

Precision in (15) is the ratio between the number of real cases without fault (true-positive) and the number of erroneously predicted cases by detection algorithm as without fault (false-positive). That metric evaluates the algorithm’s coverage, i.e., its ability to identify faults.

Recall or sensitivity in (16) is the measure of normal cases (without fault) over the number of real cases without faults. This metric allows evaluating the algorithm’s ability to detect real cases of faults. It is related to the ‘false alarm’, i.e., the fault assignment to normal cases.

Finally, f1-score in (17) is a metric that incorporates both precision and recall. It is a harmonic mean between these two metrics.
(14)$$\mathrm{Accuracy}=\frac{Q_{\mathrm{NN}}+Q_{\mathrm{FF}}}{Q_{\mathrm{NN}}+Q_{\mathrm{NF}}+Q_{\mathrm{FN}}+Q_{\mathrm{FF}}}$$
(15)$$\mathrm{Precision}=\frac{Q_{\mathrm{NN}}}{Q_{\mathrm{NN}}+Q_{\mathrm{FN}}}$$
(16)$$\mathrm{Recall}=\frac{Q_{\mathrm{NN}}}{Q_{\mathrm{NN}}+Q_{\mathrm{NF}}}$$
(17)$$F1-\mathrm{Score}=\frac{2\left(\mathrm{Precision}\times \mathrm{Recall}\right)}{\left(\mathrm{Precision}+\mathrm{Recall}\right)}$$

## 3. Short-Circuit Faults Classification

The fault classification algorithm is activated after the fault identification by the detection algorithm. The characteristics of different fault types are applied in a classification process. They are obtained from the signal samples of three-phase post-fault currents [30]. NOS and the disturbance state of each phase are determined from the properties of fault types. Thus, the fault phases are identified, and the short-circuit types are calculated via count of fault phases.

The relationships with the currents are calculated according to (18)–(20), where the maximum absolute value of currents for each phase is applied [30]:(18)$$r_{1}=\frac{\max\left\{abs\left(I_{a}\right)\right\}}{\max\left\{abs\left(I_{b}\right)\right\}}$$
(19)$$r_{2}=\frac{\max\left\{abs\left(I_{b}\right)\right\}}{\max\left\{abs\left(I_{c}\right)\right\}}$$
(20)$$r_{3}=\frac{\max\left\{abs\left(I_{c}\right)\right\}}{\max\left\{abs\left(I_{a}\right)\right\}}$$

The normalization of the relationships with the currents is performed in (21) [30]:(21)$$r_{in}=\frac{r_{i}}{\max\left\{r_{1},\;r_{2},r_{3}\right\}}\;\mathrm{with}\;i\in \left\{1,2,3\right\}$$

The behavior indices with the properties of three-phase currents are obtained via (22)–(24) [30]. The indices $C_{1}$, $C_{2}$, and $C_{3}$ contain the characteristics referring to the currents of phases A, B, and C, respectively [30]:(22)$$C_{1}=r_{1n}-r_{2n}$$
(23)$$C_{2}=r_{2n}-r_{3n}$$
(24)$$C_{3}=r_{3n}-r_{1n}$$

### 3.1. Fuzzy Inference System for Short-Circuit Faults Classification

The FIS allows us to intuitively incorporate the empirical knowledge of fault experts via fuzzy rules set. In this work, the short-circuit faults classification is performed via three FIS; therefore, an FIS is implemented for each fault type. FIS has three input variables: $phase_{a}$, $phase_{b}$, and $phase_{c}$. Each input variable is associated with three membership functions of triangular and trapezoidal types with intervals [0, 1] as shown in Figure 6a. Each input variable is represented by three linguistic variables: *low*, *medium*, and *high*. The absolute values of short-circuit behavior indices obtained in (18)–(20) are the FIS inputs.

The operating state of each phase is determined through the FIS’s outputs. The FIS contains three linguistic output variables: $Phase_{a}\;State$, $Phase_{b}\;State$, and $Phase_{c}\;State$. The FIS’ outputs are represented by two trapezoidal membership functions: Normal and Short-Circuit as shown in Figure 6b.

### 3.2. Fuzzy Inference System Rule Set

A rule set is defined for each FIS, where linguistic variables are applied to identify the fault phases. The rules are built from the absolute values of $C_{1}$, $C_{2}$, and $C_{3}$ behavior indices of currents in (18)–(20). They are calculated from of three-phase current signals measured at the substation output.

Table 2 shows a rule set to classify a single-phase short-circuit. When there is a fault in phase A, the $C_{1}$ index associated with that one phase has medium or high values. The $C_{2}$ index has low values because the phases B and C have no disturbance. Finally, the $C_{3}$ index has a negative sign, since it is obtained by subtracting between $r_{3n}$ and $r_{1n}$, and $r_{1n}$ has a high value, because phase A is shorted.

The behavior indices associated with the phases with disturbances normally present medium and high values in two-phase short-circuits, while the phases without anomaly have low values. Thus, a rule set is elaborated considering these characteristics as shown in Table 3. Finally, Table 4 shows the rule set for three-phase short-circuits, where the behavior indices have high or medium values.

### 3.3. Short-Circuit Classification

The short-circuit classification is performed after the identification of the phases with anomaly. The fault type is determined from the count of defective phases. The FIS outputs belong to the range [0, 1]. Thus, there is a post-processing step, where FIS outputs become binary via application of (25). Bit 0 indicates the normal state phase, while bit 1 indicates the shorted phase.
(25)$$S_{i}=\{\begin{array}{c} 0,\;\;Phase_{i}\;State\leq LC_{i}\;\mathrm{where}\;i\in \left\{a,b,c\right\}\\ 1,\;\;\;\;\;\;\;\;\;\;\;\;\;\;\;\;\;\;\;\;\;\;\;\;\;\;\;\;\;\;\;\;\;\;\;\;\;\;\;\;\;\;\;\;\;\;\;\;\;\;\;\;\mathrm{otherwise} \end{array}$$
where: subscript i represents the abc phases. $S_{i}$ are the FIS outputs associated with the $Phase_{i}$ and $LC_{i}$ are the limits for abc phases classification. The short-circuit types are identified using (26):(26)$$\mathrm{Short}\;\mathrm{Circuit}\;\mathrm{Type}=\{\begin{array}{l} \mathrm{single}-\mathrm{phase},\;\;\mathrm{if}\;\overset{3}{\underset{i=1}{\sum }}S_{i}=1\\ \mathrm{two}-\mathrm{phase},\;\;\;\;\;\mathrm{if}\;\overset{3}{\underset{i=1}{\sum }}S_{i}=2\\ \mathrm{three}-\mathrm{phase},\;\;if\;\overset{3}{\underset{i=1}{\sum }}S_{i}=3\\ \mathrm{Absent},\;\;\;\;\;\;\;\;\;\;\;\;\mathrm{if}\;\overset{3}{\underset{i=1}{\sum }}S_{i}=0 \end{array}$$

Figure 7 shows a flowchart with an overview of the proposed methodology, addressing the steps for detection and classification of short-circuit faults.

## 4. Results and Discussions

### 4.1. Test System Modelling IEEE-34 Bus

Different fault configurations are simulated in IEEE-34 bus test system. It consists of an extensive, lightly loaded North American real feeder with 24.9 kV nominal voltage. Its main feeder has three-phase branches, single-phase side branches, and single-phase and three-phase loads. The IEEE-34 bus main characteristics are listed as [31]: (i) two voltage regulators, (ii) an in-line transformer for voltage reduction from 24.9 kV to 4.16 kV, (iii) unbalanced distributed loads and balanced concentrated loads, and (iv) shunt capacitors.

The modified IEEE-34 bus is modeled in ATP/EMTP and its ATPDraw graphical interface. All fault combinations are simulated with 10.8 kHz sampling rate. The loads are modeled with a constant impedance model [32]. Short-circuit detection and classification algorithms are developed in MATLAB^®^ environment. All simulations are performed on a computer with an Intel Core i7 processor; 1.8–1.99 GHz and 8 GB of RAM.

Three different scenarios are modeled in the IEEE-34 bus system to evaluate the proposed methodology. Different short-circuit conditions are performed considering the variation of the parameters: fault resistance, insertion angle, short-circuit type, fault location bus, and DG units.

In the first scenario, IEEE-34 bus test system is modeled without DG units as shown in Figure 8a. In the second scenario, according to Figure 8b, two synchronous generators described in [33] are inserted at buses 840 and 848 with 220 V and 480 V of nominal voltage, respectively. Finally, Figure 8c shows the third scenario, where a PV generator is introduced at bus 848. The PV model applied in this study is based on [34] with 1 MVA of installed power.

Eleven types of simulated short-circuits are considered in several buses along the feeder. The system parameters applied for different fault scenarios are presented in Table 5. Short-circuits with resistance values greater than 40Ω are considered as HIF, where a specific model for it is introduced as shown in Figure 2.

### 4.2. Short-Circuit Fault Detection

The fault state can be determined by comparative analysis of energies and normalized mean with a variation limit for each phase. They are empirically determined from a test set, in which a discrepancy between the phase values under NOS and with disturbance is observed. The accuracy and the number of simulated cases are shown in Figure 9 for each scenario.

The input signal is evaluated at a sampling rate of 10.8 kHz which is equivalent to 180 samples per cycle. The simulation time for each short-circuit type is 0.2 s; therefore, there are 12 cycles and 2160 signals. Lower and upper limits are established for each data window to ensure the reliability of the acquired samples; thus, approximately 5% of initial and final data are removed.

The number of simulations and the accuracy for short-circuit fault detection are shown in Figure 9. Single-phase, two-phase, and three-phase short-circuits are detected for the scenarios 1, 2, and 3. Noteworthy that after the insertion of DG units, there is a reduction in accuracy.

The values obtained for normalized energy or for normalized mean of three-phase currents in some phases with NOS are high in some parameter combinations; therefore, the pre-established variation limit $LD_{i}$ is exceeded in some simulated cases. Consequently, the algorithm wrongly identifies some perturbations.

### 4.3. Short-Circuit Fault Classification

The classification process is performed after the identification of a fault by the detection algorithm. The classification consists of identifying the phases involved in short-circuit. The operational status of each phase is obtained from FIS outputs. The fault classification is obtained from the number of defective phases.

As shown in Figure 6, the input variable ${\mathrm{Phase}}_{i}\;\mathrm{Input}$ assumes one among three linguistic values represented by membership functions, where two that are trapezoidal are named “low” and “high”, and the other one is triangular, named “medium”. The parameters of these membership functions applied for short-circuit classification are shown in Table 6.

The phase operational state is obtained from the FIS outputs. It is represented by the variable ${\mathrm{Phase}}_{i}\;\mathrm{State}$ that assumes two linguistic values associated with two trapezoidal membership functions: “Normal” and “Short-Circuit”. The parameters of these membership functions are shown in Table 7.

Table 8 shows the number of simulations performed for each scenario. For each fault type, thousands of simulations are performed along the feeder for both LIF and HIF.

Figure 10a shows the results for the short-circuit classification for scenario 1. In total, 6860 simulations were carried out, with 2460, 3300, and 1100 FIS simulations for classification of single-phase, two-phase, and three-phase short-circuits, respectively. The proposed methodology presents satisfactory performance for LIF and HIF.

The results obtained for the short-circuit classification for scenario 2 with two synchronous generators DG units are shown in Figure 10b, where 6825 simulations are performed with 2425, 3300, and 1100 SIFs’ simulations for classification of single-phase, two-phase, and three-phase short-circuits, respectively.

Finally, Figure 10c shows the results for scenario 3, where DG units with PV panels are considered and 6480 simulations are performed with 2200, 3180, and 1100 SIFs’ simulations for classification of single-phase, two-phase, and three-phase short-circuits, respectively.

According to Figure 10, there is a drop in accuracy for the faults at scenarios 2 and 3, where DGs units are introduced: two synchronous generators and PV panels.

Additionally, in some parameter combinations, especially that ones with high resistances for single-phase and two-phase short-circuits, the behavior indices of normal phases present high values; therefore, they exceeded the limit established $LC_{i}$ for each phase. Consequently, the classifier erroneously identifies a normal phase as defective phase in some cases.

On the other hand, the classifications with error occur in three-phase short-circuits due to behavior index with low values for the phases with disturbance; therefore, the classifier identifies them as normal, since the tolerance limits $LC_{i}$ are not violated. It is worth noting that the same FIS set is used for all simulated scenarios (with and without DG units); therefore, there are no changes in membership functions parameters.

### 4.4. Methodology Validation for Detection and Classification of Short-Circuits Faults

This section presents metrics for evaluating the proposed methodology for detecting and classifying short-circuit faults for scenarios 2 and 3. It is noteworthy that scenario 1 has an accuracy of 100% for short-circuit fault detection and classification; therefore, it is not mentioned in this section.

#### 4.4.1. Detection Stage Validation

The short-circuit detection stage performance is evaluated in this section for scenarios 2 and 3 for single-phase, two-phase, and three-phase short-circuit faults. In this sense, cases with short-circuit absent (NOS) and short-circuit fault are simulated to obtain the confusion matrix and its metrics to evaluate the detection stage performance.

Table 9 shows the confusion matrix elements and its metrics to evaluate the detection stage performance. It is noteworthy that the accuracy and precision have high values for all cases simulated. Therefore, the detection stage has a good coverage, i.e., it identifies most short-circuit faults.

On the other hand, the recall has a lower value than the accuracy and precision metrics. Thus, in some cases, there is the fault assignment to normal cases or “false alarms”, especially for the three-phase short-circuit at scenario 3. F1-score metric incorporates the outcomes of precision and recall metrics.

The insertion of synchronous generators (scenario 2) and PV panels (scenario 3) DG units changes the short-circuit currents of IEEE 34-bus test system and, in some cases, the detection stage assigns fault in normal cases, resulting in “false alarms”.

#### 4.4.2. Classification Stage Validation

The short-circuit classification stage performance is evaluated in this section for scenarios 2 and 3 for single-phase, two-phase, and three-phase short-circuit faults. Classification stage is applied after the fault detection stage.

There is an FIS classifier for each single-phase, two-phase, and three-phase fault type. Table 10 shows the FIS classifier performance for scenarios 2 and 3. For example, the FIS classifier for single-phase faults has 2365 hits (faults classified correctly) and 94 errors (faults classified incorrectly) for scenario 2. Thus, its accuracy is 96.2%. There are 52 faults classified incorrectly as single-phase. In this sense, suppose that there is a fault at phase A; however, according to FIS classifier, the fault is in B or C phases. Therefore, that is an illustrative example of single-phase fault classified incorrectly.

It is worth pointing out that the accuracy metric has high value, and it is greater than 95.4% for all simulated cases.

## 5. Conclusions

A methodology for detection and classification of short-circuit faults in electric power distribution systems (EPDS), considering the insertion of distributed generation (DG) units, was presented in this work. The fault detection algorithm was based on continuous analysis of current signals measured at the substation output, and whose characteristics were extracted through the application of maximum overlap discrete wavelet transform (MODWT) and the mean of current signals. The system operating state was defined via comparative analysis of the normalized energy and mean values of currents, with variation limits previously established for each phase. Three fuzzy inference systems (FIS) were implemented to classify the faulty phases. The behavior indices of three-phase currents were the inputs of each FIS.

The proposed methodology was evaluated in the modified IEEE 34-bus test system and represented in ATPDraw. The proposed methodology was evaluated via 20,000 simulated cases considering three scenarios with and without DG units and considering variations of the following parameters: type of fault (single-phase, two-phase, three-phase), angle of incidence, fault resistance (high impedance fault (HIF) and low impedance fault (LIF)), fault location bus, and types of DG units (synchronous generators and photovoltaic (PV) panels).

In the fault detection stage, greater than 94.9% of short circuits were correctly detected in all scenarios with and without DG units. On the other hand, in the fault classification stage, the classifier achieved a high accuracy percentage, where 100% of short-circuits were classified correctly for scenario 1 (without GD units). In scenarios 2 and 3 (with GD units), the accuracy was greater than 95.4% for all simulated cases. The insertion of DG units increases the short-circuit current; therefore, in some cases, the classification system may erroneously indicate faults on a feeder in normal operating state (NOS).

The proposed methodology is efficient. It presented accuracy greater than 94.9% for detection and classification of all short-circuit faults, even after the insertion of DG units.

## Figures and Tables

**Figure 1 sensors-22-09418-f001:**
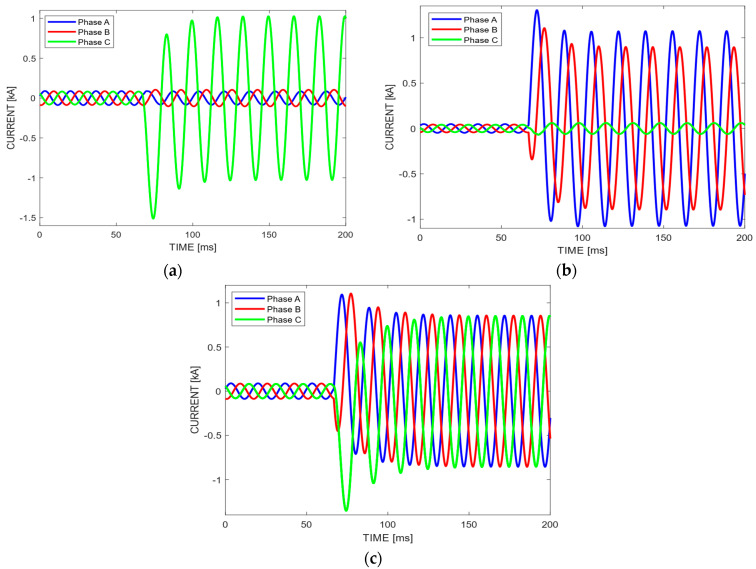
Fault types: (**a**) line-to-ground fault (SLGF); (**b**) double line-to-ground fault (DLGF); and (**c**) three-phase-ground fault (LLLGF).

**Figure 2 sensors-22-09418-f002:**
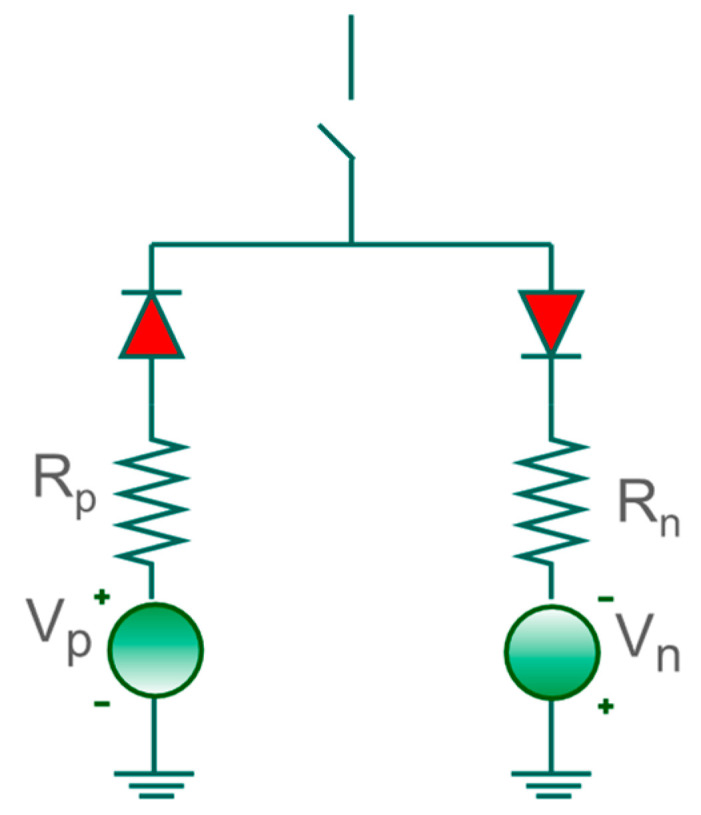
Model adopted for HIF [23].

**Figure 3 sensors-22-09418-f003:**
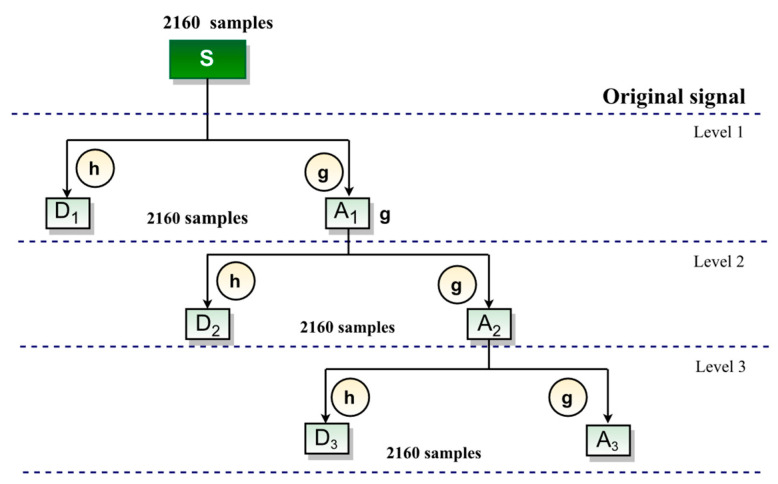
Illustrative example of MODWT filtering process [28].

**Figure 4 sensors-22-09418-f004:**
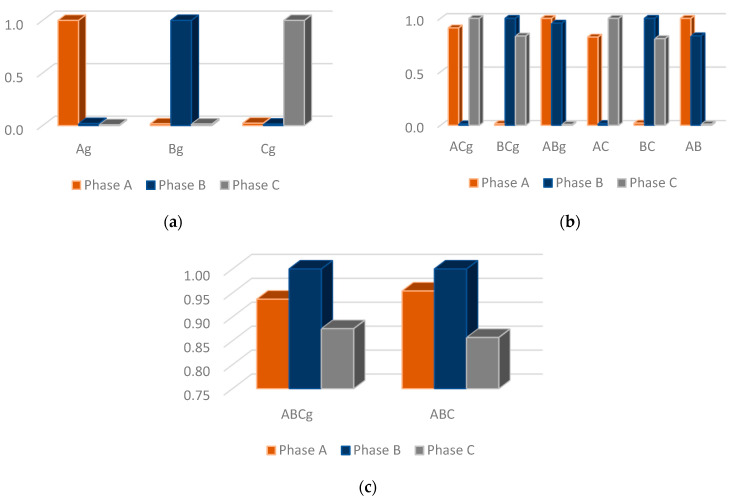
Illustrative example with the normalized energies. Short-Circuits: (**a**) single-phase; (**b**) two-phase; and (**c**) three-phase.

**Figure 5 sensors-22-09418-f005:**
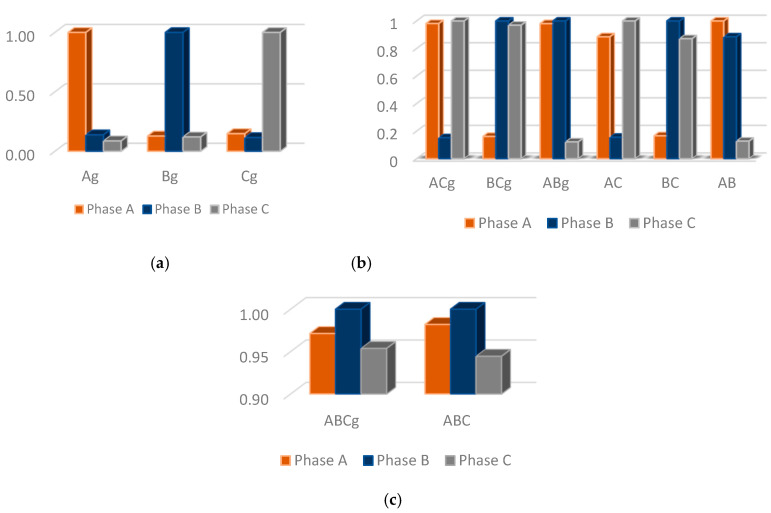
Illustrative example with the means of normalized current signals. Short-Circuits: (**a**) single-phase; (**b**) two-phase; and (**c**) three-phase.

**Figure 6 sensors-22-09418-f006:**
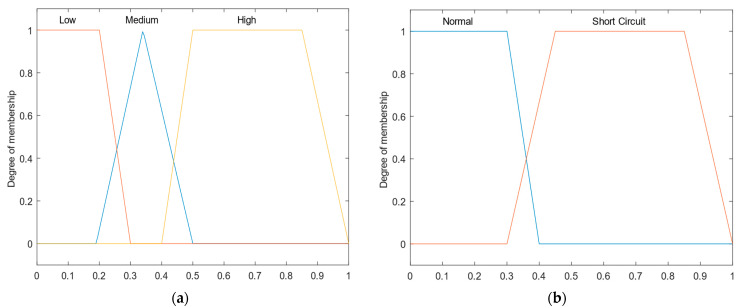
Linguistic variables and membership functions. Linguistic variables: (**a**) Input variables; and (**b**) Output variables.

**Figure 7 sensors-22-09418-f007:**
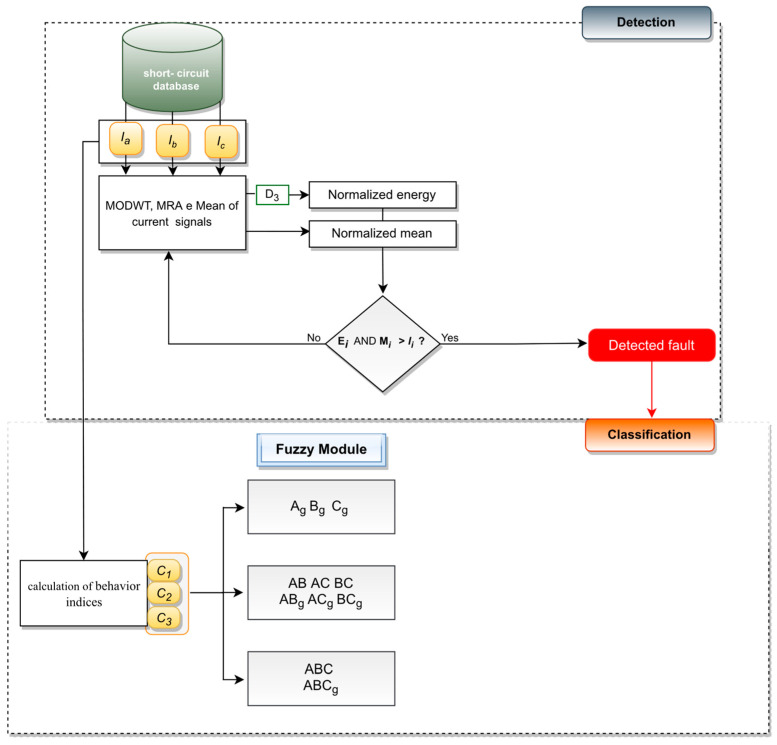
Flowchart with an overview of proposed methodology.

**Figure 8 sensors-22-09418-f008:**
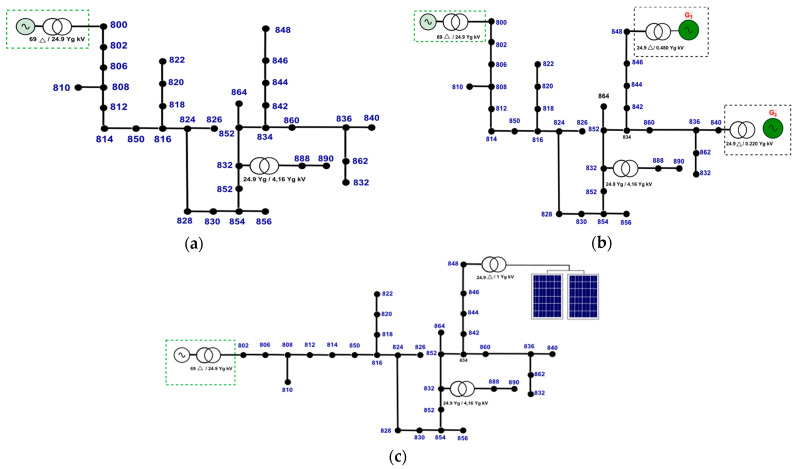
IEEE-34 bus test system. Fault Simulation: (**a**) Without DG unit (Scenario 1); (**b**) With two synchronous generators (Scenario 2); and (**c**) With PV panels (Scenario 3).

**Figure 9 sensors-22-09418-f009:**
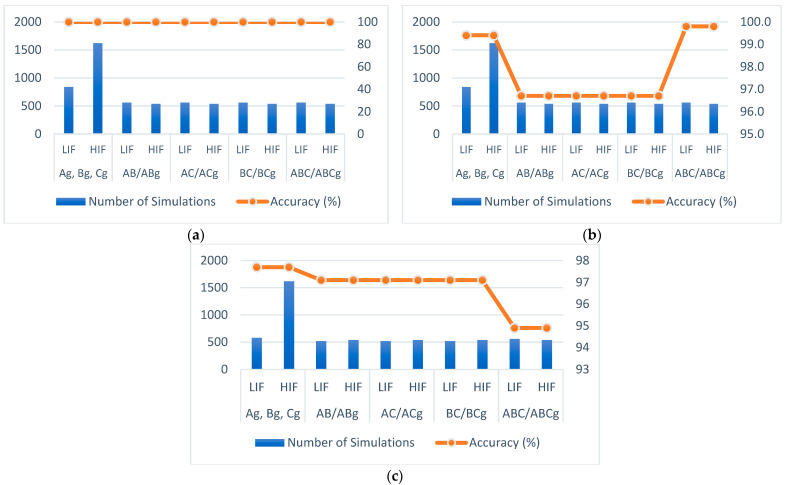
Short-circuit detection for three scenarios: (**a**) without DG units (scenario 1), (**b**) with two synchronous generators (scenario 2), and (**c**) with PV panels (scenario 3).

**Figure 10 sensors-22-09418-f010:**
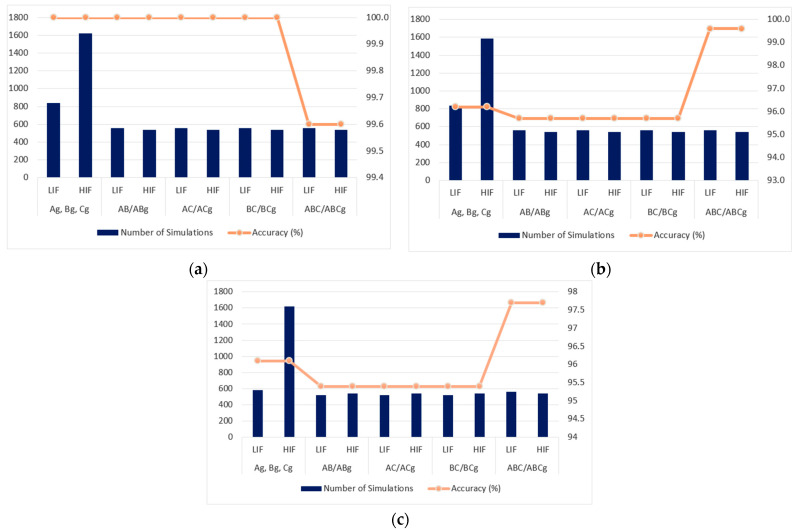
Short-circuit classification for three scenarios: (**a**) without DG units (scenario 1), (**b**) with two synchronous generators (scenario 2), and (**c**) with PV panels (scenario 3).

**Table 1 sensors-22-09418-t001:** Confusion matrix structure.

Confusion Matrix		Predicted State	
		N (Normal)	F (Fault)
Real State	N (Normal)	$Q_{NN}$	$Q_{NF}$
	F (Fault)	$Q_{FN}$	$Q_{FF}$

**Table 2 sensors-22-09418-t002:** FIS rules set to classify single-phase short-circuit.

Rules	Short-Circuit	Behavior Indices of Three-Phase Currents		
		$C_{1}$	$C_{2}$	$C_{3}$
1st	$A_{g}$	High	Low	High
2nd	$A_{g}$	Medium	Low	High
3rd	$A_{g}$	High	Low	Medium
4th	$B_{g}$	High	High	Low
5th	$B_{g}$	High	Medium	Low
6th	$B_{g}$	Medium	High	Low
7th	$C_{g}$	Low	High	High
8th	$C_{g}$	Low	Medium	High
9th	$C_{g}$	Low	High	Medium

**Table 3 sensors-22-09418-t003:** FIS rules set to classify two-phase short-circuit.

Rules	Short-Circuit	Behavior Indices of Three-Phase Currents		
		$C_{1}$	$C_{2}$	$C_{3}$
1st	$AB_{g}$	High	High	Low
2nd	$AB_{g}$	High	Medium	Low
3rd	$AB_{g}$	Medium	High	Low
4th	$AC_{g}$	High	Low	High
5th	$AC_{g}$	Medium	Low	High
6th	$AC_{g}$	High	Low	Medium
7th	$BC_{g}$	Low	High	High
8th	$BC_{g}$	Low	Medium	High
9th	$BC_{g}$	Low	High	Medium
10th	*AB*	High	High	Low
11th	*AC*	High	Low	High
12th	*BC*	Low	High	High

**Table 4 sensors-22-09418-t004:** FIS rules set to classify three-phase short-circuit.

Rules	Short-Circuit	Behavior Indices of Three-Phase Currents		
		$C_{1}$	$C_{2}$	$C_{3}$
1st	ABC	Medium	Medium	Medium
2nd	$ABC_{g}$	High	Medium	Medium
3rd	$ABC_{g}$	High	Medium	High

**Table 5 sensors-22-09418-t005:** IEEE-34 bus test system parameters for fault simulations.

Parameters	Configurations
Fault types	$A_{g}$, $B_{g}$,$\;C_{g}$, *AB*, *AC*, *BC*,$\;AB_{g}$,$\;AC_{g}$, $BC_{g}$, *ABC*, $ABC_{g}$
Fault Location bus	806, 810, 814, 824, 828, 830, 840, 850, 854, 860
DG Units	Two synchronous generators and PV panels
Fault resistance	1 Ω to 40 Ω (LIF) and 40 Ω to 300 Ω (HIF)
Fault insertion angle	0°, 30°, 45°, 60°, 90°, 120°, and 150°

**Table 6 sensors-22-09418-t006:** Membership functions parameters for FIS input variables.

Fault Types	Membership Functions Parameters		
	Low	Medium	High
Single-Phase	[0.00, 0.00, 0.1, 0.4]	[0.20, 0.34, 0.50]	[0.35, 0.50, 0.85, 1.00]
Two-Phase	[0.01, 0.10, 0.20, 0.35]	[0.3, 0.40, 0.50]	[0.40, 0.50, 0.80, 1.00]
Three-Phase	[0.00, 0.00, 0.10]	[0.00, 0.10, 0.50]	[0.20, 0.45, 0.55, 1.00]

**Table 7 sensors-22-09418-t007:** Membership functions parameters for FIS output variables.

Fault Type	Membership Functions Parameters	
	Normal	Short-Circuits
Single-phase	[0.00, 0.00, 0.10, 0.40]	[0.200, 0.400, 0.850, 1.000]
Two-phase	[0.00, 0.00, 0.30, 0.40]	[0.300, 0.400, 0.700, 1.000]
Three-phase	[0.00, 0.00, 0.10]	[0.046, 0.094, 0.600, 0.950]

**Table 8 sensors-22-09418-t008:** Number of simulations per scenario and fault type.

Fault Type	Class	Number of Simulations		
		Scenario 1	Scenario 2	Scenario 3
$A_{g}$ */* $B_{g}$ */* $C_{g}$	LIF	840	840	580
	HIF	1620	1585	1620
*AB/* $AB_{g}$	LIF	560	560	520
	HIF	540	540	540
*AC/* $AC_{g}$	LIF	560	560	520
	HIF	540	540	540
*BC/* $BC_{g}$	LIF	560	560	520
	HIF	540	540	540
*ABC/* $ABC_{g}$	LIF	560	560	560
	HIF	540	540	540
Total		6860	6825	6480

**Table 9 sensors-22-09418-t009:** Confusion matrix metrics to evaluate the detection stage for scenarios 2 and 3.

Detected Fault	Scenarios	Confusion Matrix Elements				Accuracy	Precision	Recall	F1-Score
		$Q_{NN}$	$Q_{NF}$	$Q_{FN}$	$Q_{FF}$				
Single-Phase	2	290	17	0	2591	99.4%	100.0%	94.5%	97.2%
	3	241	57	0	2200	97.7%	100.0%	80.9%	89.4%
Two-Phase	2	461	122	6	3284	96.7%	98.7%	79.1%	87.8%
	3	616	56	57	3143	97.1%	91.5%	91.7%	91.6%
Three-Phase	2	196	2	0	1100	99.8%	100.0%	99.0%	99.5%
	3	210	70	0	1100	94.9%	100.0%	75.0%	85.7%

**Table 10 sensors-22-09418-t010:** FIS classifier performance for scenarios 2 and 3.

Scenario	FIS Classifier	Hits	Errors: Faults Classified Incorrectly			Accuracy
			Single-Phase	Two-Phase	Three-Phase	
2	Single-Phase	2365	52	0	42	96.2%
	Two-Phase	3154	0	138	5	95.7%
	Three-Phase	1096	0	3	1	99.6%
3	Single-Phase	2112	52	14	20	96.1%
	Two-Phase	3050	0	122	26	95.4%
	Three-Phase	1075	1	24	0	97.7%

## Data Availability

The proposed methodology was tested in the modified IEEE-34 bus test system. Available online: https://cmte.ieee.org/pes-testfeeders/resources/ (accessed on 10 October 2021).

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
