# Peer review of "Efficient Methodology for Detection and Classification of Short-Circuit Faults in Distribution Systems with Distributed Generation"

_sensors, 2022, doi:10.3390/s22239418_

Round 1

Reviewer 1 Report

In this context, this study presents composed of two stages of a robust methodology for short-circuit faults detection and classification with distributed generation units. In the former stage, the detection is based on the continuous analysis of three-phase currents, whose characteristics are extracted through maximal overlap discrete wavelet transform. In the latter  stage, the classification is based on three fuzzy inference systems to identify the phases with disturbance. The short-circuit type is identified by counting the shorted phases. It is a well-structured paper with interesting results. However, it requires further improvements.

(1)   The abstract should be improved. Your point is your own work that should be further highlighted.

(2)   The novel of this paper is clearly inadequate.

(3)   More statistical methods are recommended to analyze the experimental results

(4)   The values of parameters could be a complicated problem itself, how the authors give the values of parameters in the used methods.

(5)  In the Section 3, why is the Fuzzy Inference System used? Authors should describe it in detail.

(6)  At Line 319, "All fault combinations are simulated with 10.8 kHz sampling rate". The sampling rate is 10.8 kHz, can the other sampling rate be used in here?

(7)  The inspiration of your work must be highlighted. I would suggest adding some recent literature in the manuscript. For example, https://doi.org/10.3934/mbe.2023090; https://doi.org/10.1016/j.ins.2022.11.019; https://doi.org/10.1016/j.ymssp.2022.109422;https://doi.org/10.1109/JSTARS.2021.3059451 and so on.

(8)  There are some grammatical errors seen in the paper. Check carefully for a few clerical errors and formatting issues.

Author Response

Dear Editor / Reviewer:

We are resubmitting the paper entitled “Efficient Methodology for Detection and Classification of Short-Circuit Faults in Distribution Systems with Distributed Generation” to the Sensors Journal to attend to the recommendations of the Reviewers We take this opportunity to thank the Reviewers immensely for their excellent contributions, which provided improvements in the quality of the writing of this article. File "Response_reviewer_1" Attached.

Yours sincerely,

Best regards.

Carlos Roberto Minussi, Ph.D., Full Professor

UNESP - São Paulo State University,

Câmpus of Ilha Solteira

Electrical Engineering Department

Av. Brasil 56

15.385-000 Ilha Solteira (SP), Brazil

Email: minussi@dee.feis.unesp.br

Reviewer 2 Report

In the article under review, the authors propose a new methodology for detecting and classifying short-circuit faults in electric power networks.

The Introduction discusses the prerequisites for conducting research and describes their background, the literature review is clearly structured and the contributions of the authors of the paper are clearly articulated. The main parts of the paper provide descriptions of faults and methods for detecting them. A fault detection algorithm proposed by the authors is presented, the efficiency of which was verified via numerous simulations in the MatlAb and ATP / EMPT (Alternative Transients Program / ElectroMagnetic Transients Program) software

In general, this is an interesting work, the results of which can be useful to specialists in the field of the electric power industry.

During the review, I drew attention to the following shortcomings and I would like to ask the authors for a few clarifications:

  1. What is the reason for choosing a set of three scenarios when modeling the system? (section 4.1, lines 329-334).
  2. The Conclusion states that the accuracy percentage for detecting and classifying short-circuit faults is 95%. What is the basis for confidence in such high accuracy for scenarios not considered in the paper?
  3. Is there a practical application of the developments proposed by the authors? Please give a few examples. Do the authors of the paper have plans for further research?

In general, the paper is well structured. All ideas and solutions are described quite clearly. I can recommend that the paper be accepted after minor revision.

Author Response

Subject:   Resubmission (attending to Reviewers) the article to the Sensors Journal

Dear Editor / Reviewer:

We are resubmitting the paper entitled “Efficient Methodology for Detection and Classification of Short-Circuit Faults in Distribution Systems with Distributed Generation” to the Sensors Journal to attend to the recommendations of the Reviewers We take this opportunity to thank the Reviewers immensely for their excellent contributions, which provided improvements in the quality of the writing of this article. File "Response_reviewer_2" Attached.

Yours sincerely,

Best regards.

Carlos Roberto Minussi, Ph.D., Full Professor

UNESP - São Paulo State University,

Câmpus of Ilha Solteira

Electrical Engineering Department

Av. Brasil 56

15.385-000 Ilha Solteira (SP), Brazil

Email: minussi@dee.feis.unesp.br

Round 2

Reviewer 1 Report

This paper can be accepted now.